# Patient-Derived Tumor Organoids to Model Cancer Cell Plasticity and Overcome Therapeutic Resistance

**DOI:** 10.3390/cells14181464

**Published:** 2025-09-18

**Authors:** Roberto Coppo, Masahiro Inoue

**Affiliations:** Department of Clinical Bio-Resource Research and Development, Graduate School of Medicine, Kyoto University, Kyoto 606-8501, Japan; masa_inoue@kuhp.kyoto-u.ac.jp

**Keywords:** patient-derived tumor organoids, cellular plasticity, cancer stem cells, drug-tolerant persister cells, tumor heterogeneity, therapy resistance

## Abstract

Cancer cell plasticity, defined as the ability of tumor cells to reversibly adopt distinct functional states, plays a central role in tumor heterogeneity, therapy resistance, and disease relapse. This process enables cells to enter stem-like, dormant, or drug-tolerant persister states in response to treatment or environmental stress without undergoing genetic changes. Such reversible transitions complicate and limit current treatments. Conventional cancer models often fail to capture the complexities of these adaptive states. In contrast, patient-derived tumor organoids (PDOs), which retain the cellular diversity and structure of primary tumors, provide a unique system for investigating plasticity. This review describes how PDOs can model cellular plasticity, such as the emergence of drug-tolerant persister cells and the interconversion between cancer stem cell states across multiple tumor types. We particularly focused on colorectal cancer organoids, for which research on the mechanism of plasticity is the most advanced. Combined with single-cell analysis, lineage tracing, and functional assays, PDOs can help identify the molecular pathways that control plasticity. Understanding these mechanisms is important for developing therapies to prevent treatment failure and control disease progression.

## 1. Introduction

Despite major advances in early detection and treatment, cancer remains a leading cause of mortality worldwide [1]. Cancer cell plasticity, the reversible transition between distinct phenotypic states, has emerged as a critical mechanism underlying intratumor heterogeneity, therapeutic resistance, and recurrence [2,3]. This non-genetic adaptability facilitates the emergence of cancer stem-like and drug-tolerant persister (DTP) cells that survive treatment and contribute to tumor regrowth [4]. Understanding the regulation of these dynamic states is essential to improve cancer therapies.

The development of effective treatments is limited by the absence of reliable models that accurately replicate the complexity of human tumors. Although traditional 2D cancer cell lines are easy to manipulate and scale, they fail to capture the cell diversity and phenotypic heterogeneity of primary tumors. These models often accumulate significant genetic alterations over a high number of passages and lack tissue-specific niche interactions that influence tumor behavior and therapeutic responses [5]. Genetically engineered mouse models and patient-derived xenografts offer improved in vivo relevance by incorporating tumor–stroma interactions, vascularization, and host immune responses; however, they are limited by high costs, technical complexity, and lack of throughput [6,7].

In recent years, organoid cultures have emerged as a powerful alternative for modeling human cancers. Organoids are three-dimensional epithelial structures derived from either adult or pluripotent stem cells, and they can be established from both normal and tumor tissues. These cultures maintain key histological, genomic, and functional features of the tissue of origin, including cell heterogeneity, cell lineage architecture, and dependence on niche factors, such as Wnt, R-spondin, EGF, and Noggin, which normally sustain epithelial stem cells [8]. Patient-derived tumor organoids (PDOs) have demonstrated remarkable utility in modeling cancer cell dynamics. When combined with genome editing, lineage tracing, or co-culture, they offer a powerful platform for investigating the mechanisms underlying tumor initiation, progression, and resistance [9,10,11].

Previous review articles have mainly outlined the potential of organoid-based cancer modeling and its clinical applications [9,10,11,12]. The present review focuses on organoid systems as a platform for investigating cellular plasticity and summarizes research findings on the dynamics of treatment resistance using PDO, with particular attention paid to cancer stem cell (CSC) and drug-tolerant persister (DTP) populations and strategies to overcome plasticity-driven resistance.

## 2. Cancer Cell Plasticity Overview

Cell plasticity refers to the ability of cells to reprogram and adopt alternative fates or identities to support tissue regeneration and homeostasis. Traditionally, plasticity has been observed in physiological contexts, such as development and tissue repair, where it enables cells to respond dynamically to injury or environmental changes [13]. For instance, a strict unidirectional hierarchy exists in the normal colonic epithelium wherein LGR5-positive intestinal stem cells (ISCs) give rise to transient amplifying (TA) cells, which then differentiate into absorptive, secretory, and neuroendocrine lineages to maintain homeostasis [14]. However, under conditions of severe tissue damage, even committed differentiated cells such as Paneth cells can undergo dedifferentiation to replenish the ISC pool [15], demonstrating that plasticity can also emerge as an adaptive regenerative response.

Beyond natural contexts, artificially induced plasticity has been demonstrated through reprogramming technologies, with the most notable development being the generation of induced pluripotent stem (iPS) cells from terminally differentiated somatic cells, such as fibroblasts [16]. Although this process does not occur naturally, it reveals the profound potential of transcription factor–mediated reprogramming to reverse cell fate, challenging the notion of fixed cellular identity. Once considered unidirectional and irreversible, lineage specification and differentiation are now recognized as dynamic processes, with cell identity exhibiting substantial plasticity as cells adopt alternative fates in response to internal or external cues [17].

In pathological contexts such as cancer, cell plasticity enables dynamic and reversible transitions between phenotypic states in response to selective pressures such as microenvironmental cues or therapeutic interventions [3,18,19]. Plasticity can manifest through dedifferentiation (the return to an undifferentiated state along the same lineage), transdifferentiation (switching from one differentiated cell type to another, as seen in metaplasia) [20] (Figure 1), and epithelial-to-mesenchymal transition (EMT), in which epithelial cells acquire mesenchymal traits and lose polarity and adhesion [21]. In addition to these classical forms, cell plasticity encompasses reversible transitions between cancer stem–like and non-stem-like states [8] and between distinct stem cell states [22]. Furthermore, plasticity underlies the ability of tumors to shift between growth states, such as between slow-cycling DTP cells and fast-proliferating tumor cells [4], enabling evasion of targeted therapies and fueling relapse (Figure 1).

This plastic behavior is the central mechanism by which tumors acquire functional diversity, evade targeted therapies, and adapt to new microenvironments during progression and metastasis [3,18].

### 2.1. From Homeostasis to Tumorigenesis

In adult tissues, cellular homeostasis is maintained by multipotent or lineage-restricted stem cells that sustain differentiated populations. During tissue injury or regeneration, committed progenitor cells exhibit enhanced self-renewal, expanding their differentiation potential to compensate for cellular loss and support tissue repair [38]. The intestinal epithelium exemplifies this adaptability where, under normal conditions, renewal is driven by LGR5^+^ stem and transit-amplifying cells [39,40]. However, upon injury, more committed cells, including BMI1^+^ progenitors, ALPI^+^ enterocyte precursors, DLL1^+^ secretory progenitors, and Paneth cells, can revert to a stem-like state and re-establish the LGR5^+^ pool [41,42,43,44]. Transdifferentiation has also been observed under normal physiological conditions. In the skin, basal and hair follicle stem cells contribute to wound repair by reprogramming into interfollicular epidermal stem cells under the influence of a niche [45]. Similarly, in glandular epithelia, such as the mammary gland and prostate, basal cells can reacquire multipotency and regenerate luminal lineages when they are lost or ablated [46,47,48,49]. EMT also plays a key role in embryonic development and wound healing by enabling epithelial cells to transiently adopt mesenchymal traits that facilitate migration and tissue repair [21].

This cellular plasticity has significant implications in tumorigenesis. Differentiated cells can reacquire stem-like properties under the influence of oncogenic drivers, whereas tumor suppressors such as TP53 and PTEN normally restrict this potential [20]. Environmental signals further modulate this plasticity. For example, in the mouse intestine, inflammatory cues lead to the loss of LGR5^+^ cells, inducing Paneth cells to adopt a stem-like phenotype and support regeneration. In the absence of inflammation, only stem cells formed tumors after APC deletion. However, when APC loss was combined with NFKBIA deletion, constitutive NF-κB activation mimics inflammatory signaling, enabling non-stem cells to acquire tumor-initiating potential [50,51,52]. These findings highlight how the microenvironment-associated inflammatory pathways can reprogram cell fate and expand the pool of tumor-initiating cells.

### 2.2. CSCs: From a Static to a Dynamic Model

CSCs are characterized by their ability for self-renewal and regeneration of tumor heterogeneity. However, CSCs are not a fixed entity. Tumor cell populations exhibit bidirectional transition between CSC and non-CSC states, enabling dynamic maintenance of the CSC pool [8]. In colorectal cancer (CRC), lineage tracing and ablation experiments demonstrated that even upon elimination of LGR5^+^ CSCs, LGR5^−^ tumor cells could regenerate the CSC pool, emphasizing that CSC identity is plastic and regulated by both intrinsic and extrinsic cues [23,24].

Additionally, a second level of cellular plasticity exists within the CSC pool itself, as CSCs do not represent a homogeneous entity, but instead comprise distinct functional states that interconvert depending on environmental conditions [22]. For example, recent studies have described two key colonic stem cell populations: proliferative (proCSCs) and revival (revCSCs) [32,53,54,55]. proCSCs are highly proliferative, LGR5^+^, OLFM4^+^, and LRIG1^+^ drive tumor expansion, whereas revCSCs, also referred to as plastic persisters, are slow-cycling, CLU^+^, drug-resistant, and prioritize survival over proliferation. Notably, revCSCs can revert to proCSCs following therapy cessation, thereby driving tumor relapse [22,32].

Furthermore, CSC plasticity is also influenced by the tumor microenvironment (TME). Niche-derived signals, such as WNT, NOTCH, and EGFR pathway activity, can actively promote or suppress transitions between CSC and non-CSC states or between distinct CSC subtypes. Stromal cues, inflammation, and other extrinsic factors serve as key modulators of CSC identity and plasticity, enabling their dynamic adaptation to environmental changes [8,56].

### 2.3. Drug-Tolerant Persister Cells and Post-Treatment Regrowth

Another manifestation of cancer cell plasticity is the emergence of DTP cells, a slow-cycling reversible state that allows cancer cells to survive therapy without acquiring genetic resistance. These transient cells arise in initially responsive tumors via an adaptive response to drug exposure [57]. Although some may originate from rare preexisting subpopulations [25,26,58,59], evidence from lineage tracing, barcoding, and mathematical modeling suggests that most DTPs are induced by therapy [60,61,62,63,64,65,66]. Characterized by transcriptional and epigenetic reprogramming, metabolic adaptation, and immune evasion, DTPs persist under treatment and retain the ability to reenter the cell cycle upon cessation of therapy, thereby enabling tumor relapse [4,57]. Under continuous therapeutic pressure, these cells may revert to a drug-sensitive state or evolve into genetically resistant clones [63,66,67,68]. Despite growing interest, the molecular basis of DTP biology remains unclear.

## 3. Organoids to Model Cellular Plasticity, Cell Fate, and Cell-State Transitions

PDOs capture tumor complexity and cellular plasticity, enabling in vitro modeling of lineage hierarchies and cell-state dynamics. Organoids provide a unique platform for modeling the interplay between tumor-intrinsic plasticity programs and extrinsic microenvironmental signals. When integrated with single-cell transcriptomics, CRISPR editing, drug perturbation assays, and in vivo transplantation, PDOs allow researchers to track, manipulate, and functionally test plastic cancer cell states, which are essential to develop a better understanding of disease progression, relapse, and mechanisms of therapeutic resistance (Table 1 and Appendix A and Figure 1).

### 3.1. Reversible Interconvertion

Reversible interconversion describes the bidirectional switching between distinct cellular compartments. These include (i) transitions between CSC and non-CSC states [23,24] and (ii) reversible shifts between growth states, such as the ability of slow-cycling DTP cells to return to their proliferative state after treatment withdrawal [25,26,27].

A study that used PDOs derived from breast cancer provided evidence that luminal-type breast cancer cells exhibit high plasticity, with their differentiation status being highly dynamic during in vitro culturing [28]. The key finding was that estrogen receptor (ER) expression, a hallmark of luminal-type breast cancer, typically diminishes after passaging of organoids, concurrent with a phenotypic shift from luminal to basal-like characteristics. This transition is associated with activation of the NOTCH signaling pathway, as indicated by increased HES1 expression and gene expression profiling showing the upregulation of NOTCH target genes. At the molecular level, activation of NOTCH signaling appears to drive dedifferentiation and stemness, promoting a cell-state transition from the luminal to a basal-like phenotype. Pharmacological inhibition of NOTCH signaling with DAPT effectively restores ER expression and reverts cells to a luminal state, suggesting that NOTCH acts as a key regulator of cellular plasticity in this context.

Pancreatic ductal adenocarcinoma (PDAC) organoids further underscore the potential of the PDO system [29,30,31]. Seino et al. [29] used a library of 39 patient-derived PDAC organoids to functionally categorize cellular states based on their dependency on stem cell niche factors. They identified three distinct subtypes: Wnt-non-secreting (W−), which depends on exogenous Wnt ligands from cancer-associated fibroblasts (CAFs); Wnt-secreting (W+), which gains niche autonomy via endogenous expression of epithelial Wnt ligands, such as WNT7B and WNT10A; and Wnt/R-spondin-independent (WRi), which is fully niche-independent. These transitions were not driven by canonical Wnt pathway mutations but were instead linked to transcriptional reprogramming regulated by GATA6. GATA6 loss, mediated by promoter hypermethylation, enabled Wnt ligand expression and a switch toward a basal-like transcriptional identity. CRISPR-mediated GATA6 knockout in W− organoids induced Wnt-autonomous growth, whereas GATA6 overexpression reversed this phenotype. Moreover, PDAC organoids engineered with KRAS, TP53, SMAD4, and CDKN2A mutations transitioned to Wnt independence only after prolonged culture, indicating that epigenetic adaptation, rather than genetic alteration, regulates this niche escape. These findings highlight a paradigm in which tumor progression involves plastic reprogramming of niche dependence, coordinated through transcriptional and epigenetic changes rather than fixed mutations.

Using an intraductal xenograft model, Miyabayashi et al. [30] transplanted patient-derived PDAC organoids into mouse pancreatic ducts and identified two progression phenotypes: indolent, classical/progenitor-like and aggressive, basal-like/squamous tumors. Transcriptomic analyses showed that these identities were not fixed but modulated by the microenvironment. Organoids classified as classical in vitro produced basal-like lesions in vivo, and lineage tracing revealed that single clones could generate both classical and invasive regions, supporting plasticity over clonal selection. KRAS hyperactivation induced a basal-like program with EMT, MYC/E2F, and ECM remodeling genes, whereas classical intraductal lesions retained GATA6 and exhibited oxidative and lipid metabolic signatures. These findings demonstrate that subtype identity in PDAC is shaped by KRAS signaling and the niche context, with transcriptional heterogeneity arising from dynamic plastic responses.

Another example was provided by Raghavan et al. [31] who used single-cell RNA sequencing of metastatic PDAC biopsies and matched PDOs to study transcriptional plasticity and its regulation by the TME. Although PDOs preserved genomic alterations, they lacked the full range of transcriptional states observed in vivo. Three malignant states were identified: classical, basal-like, and intermediate co-expressor (IC) state marked by RAS signaling, EMT, and interferon responses. These states were dynamic and modulated by extrinsic cues such as TGF-β, which promoted transitions toward IC and basal-like phenotypes. These shifts were replicated in PDOs by altering growth factors and cytokines. This study highlighted how non-genetic regulation of the cell state shapes cellular behavior and demonstrated the utility of PDOs in modeling plasticity and state transitions.

Organoid-based models have advanced the understanding of cellular plasticity, stem cell hierarchies, and DTP cells in CRC (Table 1 and Figure 1). When LGR5^+^ CSCs were ablated via CRISPR-based strategies in CRC organoids, tumors initially regressed; however, they subsequently regrew owing to lineage plasticity, as LGR5^−^ differentiated cells reacquired stemness and regenerated the LGR5^+^ pool [23,24]. Using genetically engineered mouse models and organoid systems carrying mutations in *APC*, *KRAS*, *TRP53*, and *SMAD4* to mimic stepwise CRC progression, de Sousa e Melo et al. [23] demonstrated that ablation of LGR5^+^ cells effectively restricts primary tumor growth but does not cause tumor regression. Upon ablation, tumors persist via LGR5^−^ cells that exhibit stem cell plasticity, capable of regenerating the LGR5^+^ pool and sustaining tumor growth. Importantly, LGR5^+^ cells are necessary for metastasis. Notably, LGR5^−^ cells can reacquire a stem-like identity in permissive tissue-specific niches such as the colon, but not in less supportive environments such as the liver, exhibiting tissue-specific plasticity. These results support a model in which tumor cells undergo dynamic cell-state transitions between LGR5^+^ and LGR5^−^, contributing to tumor heterogeneity, progression, and metastatic potential, emphasizing the importance of targeting dynamic CSC states for effective therapeutic strategies.

Similar findings were reported by Shimokawa et al. [24], who provided direct evidence of CSC plasticity in human CRC using CRISPR-based lineage tracing and ablation in PDOs. LGR5^+^ cells showed long-term self-renewal and multilineage differentiation, confirming their CSC identity. Inducible ablation of LGR5^+^ cells caused tumor regression; however, LGR5^−^/KRT20^+^ cells later re-expressed LGR5 and regenerated the CSC pool, demonstrating bidirectional plasticity. This reprogramming was driven by niche cues and intrinsic mechanisms. The study also showed that combining targeted stem cell ablation with chemotherapy enhanced efficacy by disrupting this plasticity. Together, these findings reveal a dynamic and reversible hierarchy within CRC, in which differentiated cells retain latent regenerative potential, and demonstrate how PDOs can faithfully model these transitions.

In metastatic CRC models, liver lesions were initially seeded by LGR5^−^ cells; however, full metastatic outgrowth required re-emergence of LGR5^+^ CSCs, highlighting their role in dormancy escape and progression [36]. Using organoid-based models with Apc, Kras, Trp53, and Smad4 mutations, the study showed that LGR5^−^ cells, despite their differentiated state, could reacquire stem-like features and regenerate LGR5^+^ CSCs at metastatic sites, independent of classical niche signals (Wnt, R-spondin, and EGF). This intrinsic plasticity was enhanced by HGF and FGF. Ablation of emergent LGR5^+^ cells impaired metastatic growth, confirming that CSC reprogramming, not static hierarchies, drives metastasis. Organoid-based CRC models identified high-relapse cells (HRCs), an EMP1^+^KRT20^−^ epithelial population that drives metastatic recurrence via plasticity [37]. HRCs are found in invasion fronts and micrometastases and lack EMT-TFs, but they express partial EMT markers (LAMC2, ITGA2, and PLAUR) and revert into LGR5^+^ cells during outgrowth. Ablation of HRCs prevented relapse without affecting the primary tumors. Anti-PD1/CTLA4 immunotherapy eliminated residual HRCs, highlighting transient immune vulnerability and underscoring non-stem cell plasticity as a potential therapeutic target.

A recent study using an inducible lineage-tracing system in CRC organoid xenografts has revealed that dormant LGR5^+^p27^+^ cells, anchored at the COL17A1-enriched matrix interface, persist in a drug-tolerant state during chemotherapy and later reinitiate tumor growth [25]. COL17A1 maintains dormancy by suppressing FAK–YAP signaling. Under chemotherapeutic stress, downregulation or disruption of COL17A1 leads to the activation of FAK phosphorylation and subsequent nuclear translocation of YAP, which promotes cell cycle re-entry and tumor regrowth. Importantly, inhibition of YAP signaling effectively prevents dormant CSCs from reactivating, thereby delaying or abrogating tumor recurrence. This mechanistic insight highlights the role of cell–matrix interactions and mechanotransduction in regulating plasticity and dormancy.

Other slow-cycling populations characterized by MEX3A expression suppress Wnt signaling to evade chemotherapy and dominate the residual tumor mass post-treatment. Using PDOs and genetically engineered mouse tumor organoids, Alvarez-Varela et al. [26] demonstrated that a subset of LGR5^+^ CRC cells expressing the RNA-binding protein MEX3A adopts a latent chemoresistant state under suboptimal niche conditions (e.g., TGF-β exposure or EGF deprivation). These MEX3A^+^ cells, initially characterized by reduced proliferation, metabolic downregulation, and distinct transcriptional signatures, persist through chemotherapy and later regenerate the tumor mass. Lineage-tracing and single-cell transcriptomics revealed that upon drug exposure, MEX3A^+^ cells downregulate canonical WNT and intestinal stem cell programs and transiently transition into a YAP-dependent fetal-like state resembling revival stem cells [54,55]. This adaptive cell state switch enables survival under therapeutic pressure, and upon treatment withdrawal, reversion to the original LGR5^+^ phenotype occurs, thereby reconstituting the tumor. Notably, MEX3A deletion impaired this plastic response, promoting differentiation toward a secretory goblet-like lineage and sensitizing tumors to chemotherapy. This study demonstrated how organoid models recapitulate the dynamic, reversible state transitions underlying DTP cell biology [26].

Coppo et al. [27] revealed a novel mechanism of cancer cell plasticity using CRC PDOs by identifying two phenotypically distinct yet interchangeable cell subpopulations. This study demonstrated that CRC cells are phenotypically heterogeneous and capable of transitioning between distinct growth states and cell fates. Using a modified spheroid-formation assay, which measures the ability of a single cell to proliferate and form a spheroid, the single cell-derived spheroid-forming and growth (SSFG) assay, the researchers identified two distinct behaviors of cancer cells: cells consistently formed small spheroids (designated “S-cells”), whereas others formed larger spheroids (designated “L-cells”). S-cells are characterized by their slow-growing capacity, drug resistance, and consistent generation of only small spheroids in subsequent SSFG assays, a pattern termed the “S-pattern.” In contrast, L-cells display a high proliferative capacity, sensitivity to chemotherapy, and yield both small and large spheroids, termed the “D-pattern” phenotype. Importantly, they found that direct cell–cell contact induces the transition from S-cells to L-cells, a process that is dependent on Notch signaling, as evidenced by the inhibition of this transition with the Notch inhibitor DAPT. Notch pathway activation, as evidenced by increased HES1 expression, appears to serve as a molecular switch facilitating cell-state transitions, thereby enabling slow-growing resistant cells to acquire proliferative and drug-sensitive characteristics. Furthermore, gene expression analyses revealed that the RNA-binding protein Musashi-1 (MSI1) was significantly upregulated in L-cells. Functional studies demonstrated that MSI1 knockout impaired the conversion of S-cells to L-cells, whereas MSI1 overexpression in S-cells promoted the acquisition of the D-pattern phenotype. In vivo, S-cells acquired L-cell characteristics in response to microenvironmental cues. A similar study by Lin et al. [69] showed that CRC cell clusters newly formed from PDOs adopt distinct growth patterns—actively growing (AG) or poorly growing (PG)—that are regulated by Notch signaling upon detachment. Notch activation promotes survival and the AG fate via NICD-mediated upregulation of HES1 and CDKN1A, whereas its inhibition increases the number of PG clusters, which grow slowly and resist chemotherapy. Collectively, these findings revealed a reversible, non-genetic plasticity mechanism in CRC driven by NOTCH signaling. Through these pathways, CRC cells can switch between proliferative, drug-sensitive states and slow-growing, drug-resistant states, showing how PDO models capture phenotypic transitions that contribute to tumor heterogeneity and therapy resistance, highlighting the therapeutic potential of targeting plasticity-regulating signals.

Organoid models have advanced the understanding of cellular plasticity in early-onset CRC (EOCRC). Yan et al. [70] created a biobank of EOCRC organoids by identifying APC-wild-type tumors with PTPRK–RSPO3 fusions or RNF43 mutations. The RSPO fusion organoids resembled the normal epithelium and differentiated upon Wnt withdrawal, whereas APC-mutant organoids remained undifferentiated and expressed PTK7. Frequent co-mutations in SMAD4 or BMPR1A suggested that the disruption of the BMP pathway promotes plasticity. These findings highlighted how organoids reveal the impact of distinct mutations on CRC cell-state regulation.

### 3.2. Cell-State Transition

Cell-state transition refers to phenotypic shifts within the CSC pool. Cortina et al. [33] used CRISPR/Cas9 to engineer lineage-tracing reporters at the LGR5 and KI67 loci in CRC PDOs, enabling tracking of stem-like and proliferative states. In xenografts derived from these organoids, LGR5^+^ cells efficiently initiated tumors and yielded differentiated progeny marked by KRT20 and MUC2. Lineage tracing showed that LGR5^+^ cells undergo self-renewal and sustain tumor growth, with a subset in a quiescent KI67^−^ state. Transcriptomic profiling confirmed these LGR5^+^/KI67^−^ cells retained stemness markers (LGR5 and SMOC2) while downregulating cell cycle genes (e.g., KI67, AURKB, and FOXM1). This study demonstrates the functional heterogeneity and plasticity of CRC-initiating cells.

Recent high-dimensional analyses integrating single-cell transcriptomics, phospho-signaling, and computational phenoscapes have provided a detailed map of CRC plasticity [32]. These studies showed that stem cell fate in murine colonic CRC organoids spans a continuum between proliferative and revival-like states, shaped by both intrinsic oncogenic mutations and extrinsic stromal signals. Co-culture with primary fibroblasts revealed that fibroblast-derived TGF-β1 activates YAP signaling, which, together with WNT3A, induces a revival stem cell (revCSC) phenotype characterized by CLU expression, slow cycling, and chemoresistance. Conversely, organoids with APC loss and KRAS^G12D mutations hyperactivate WNT/β-catenin, MAPK, and PI3K pathways, bypassing stromal regulation and locking cells into hyperproliferative states (proCSCs) marked by LRIG1. Notably, revCSCs can revert to proCSCs after therapy withdrawal, driving tumor relapse [22,32,53]. These findings highlight how oncogenic mutations can reprogram normal epithelial plasticity by disrupting microenvironmental control, thereby favoring tumorigenic stem cell states.

### 3.3. Transdifferentiation

Transdifferentiation refers to the switch from one differentiated cell type to another [20]. A striking example was recently demonstrated in mixed small-cell neuroendocrine carcinoma (SCNEC) of the uterine cervix. Using a PDO model, Masuda et al. [34] provided functional evidence that the distinct tumor components, SCNEC and adenocarcinoma, can originate from a common progenitor cell. Single-cell transcriptomic profiling of organoids revealed the presence of three cellular states: SCNEC-like, adenocarcinoma-like, and an undifferentiated cluster lacking clear lineage markers. Importantly, HPV18 E6 oncogene expression and TP53 pathway activity varied across clusters, suggesting that viral oncogene dosage may contribute to lineage specification. Furthermore, chemotherapy induced a phenotypic shift toward an adenocarcinoma-like state, as confirmed using a KRT19 fluorescent reporter system. These observations highlight the dynamic and reversible nature of lineage commitment in cervical SCNEC and suggest that cellular plasticity, in addition to genetic divergence, drives intratumor heterogeneity and therapy resistance.

Similarly, Calandrini et al. [71] established a pediatric kidney cancer organoid biobank that preserved the histological and molecular heterogeneity of the original tumors. In Wilms tumor organoids, the triphasic composition of epithelial, stromal, and blastemal-like cells, was maintained in vitro. Single-cell RNA sequencing identified distinct populations marked by EPCAM and CDH1 (epithelial), collagen genes and THY1 (stromal), and a third population co-expressing epithelial and stromal markers along with progenitor genes such as *SIX1* and *NCAM1*, likely corresponding to blastemal-like cells. These transcriptional profiles persisted across early and late passages. Targeted sequencing confirmed the tumor origin of all cell types via shared *WT1* mutations. The presence of multiple lineage programs and transitional states reflects cell-state plasticity and provides a model for studying lineage reprogramming, tumor progression, and therapy resistance in pediatric cancers.

Togasaki et al. [35] performed studies on diffuse-type gastric cancer (GC) and used PDOs to show that signet-ring cell carcinoma (SRCC) and poorly cohesive carcinoma not otherwise specified (PCC-NOS) are not clonally distinct, but instead represent reversible differentiation states modulated by Wnt signaling. Withdrawal of Wnt and R-spondin (WR) niche factors induced a shift from a proliferative LGR5^+^ PCC-NOS state to a mucin-rich differentiated SRCC phenotype. Reintroduction of WR enabled only TP53-mutant organoids to revert to the PCC-NOS state, highlighting how genetic alterations such as TP53 loss enhance plasticity. Furthermore, stromal RSPO3^+^ fibroblasts were found to create microenvironmental niches that sustain Wnt activity and histologic identity. This study illustrates how tumor–stroma interactions and niche signals govern intratumor heterogeneity and plasticity in human gastric cancers.

**Table 1 cells-14-01464-t001:** Organoids to model cellular plasticity.

Tumor Type	Organoid Source	Key Plasticity Features	Pathway and/or Phenotype	Reference
SCNEC	SCNEC/adenocarcinoma	Lineage switch between SCNEC and adenocarcinoma	HPV18 E6 dosage, TP53 activity	[34]
Breast Cancer	Luminal-type breast cancer	Luminal to basal-like transition upon passaging	NOTCH signaling (HES1); reversible with DAPT	[28]
Wilms Tumor (Kidney)	Pediatric kidney tumor	Triphasic histology retained; mixed lineage states	SIX1, NCAM1; WT1 mutation confirms tumor origin	[71]
Gastric Cancer (GC)	Diffuse-type GC	SRCC and PCC-NOS interconvert	Wnt/RSPO withdrawal; TP53 loss; stromal RSPO3	[35]
PDAC	PDAC biobank	Wnt dependency defines cell states	GATA6 methylation regulates Wnt ligand expression	[29]
	PDAC xenograft model	Classical to basal-like switch in vivo	KRAS hyperactivation, EMT, MYC	[30]
	Metastatic PDAC PDOs	Co-expressor (IC) state modulates drug response	TGF-β, MAPK inhibitors; non-genetic transitions	[31]
Colorectal Cancer (CRC)	Genetically engineered murine colonoids	LGR5^−^ to LGR5^+^ CSC reacquisition	Niche-dependent reprogramming	[23,24]
	CRC PDOs and genetically engineered murine colonoids	Dormant LGR5^+^p27^+^ cells drive regrowth	COL17A1-FAK-YAP axis	[25]
	CRC PDOs and genetically engineered murine colonoid	MEX3A^+^ slow-cycling chemoresistant DTP state; drive tumor regrowth	YAP activation; fetal-like state	[26]
	CRC PDOs	S-to-L cell transition (S-pattern vs. D-pattern)	Notch signaling, MSI1 upregulation	[27]
	Genetically engineered murine colonoid	revCSC-to-proCSC transitions	TGF-β, YAP, WNT3A vs. KRAS/APC mutations	[32]
	CRC PDOs	LGR5^+^/KI67^−^ quiescent CSCs	Stem-like but non-proliferative	[33]
	Genetically engineered murine colonoids	Metastasis initiated by LGR5^−^, outgrowth needs LGR5^+^	HGF/FGF signals promote stem reacquisition	[36]
	Orthotopic CRC models	HRCs drive metastasis relapse	EMP1^+^, immune vulnerability window	[37]
	CRC PDO clusters	AG vs. PG cluster formation at detachment	Notch cleavage, HES1, CDKN1A	[69]
	Early-onset CRC PDOs	RSPO fusions retain differentiation	PTPRK-RSPO3, SMAD4/BMPR1A loss	[70]

SCNEC, small-cell neuroendocrine carcinoma; PDAC, pancreatic ductal adenocarcinoma; PDO, patient-derived tumor organoid; CSC, cancer stem cell; DTP, drug-tolerant persister cell; S-cell, slow-growing cell; L-cell, fast-growing cell; S-pattern, slow-growing pattern; D-pattern, dual (slow and fast)-growing pattern; AG, actively growing; PG, poorly growing; revCSC, revival stem cell; proCSC, proliferative stem cell; HRC, high-relapse cell.

## 4. Rethinking Cancer Treatment: How to Target a Moving System?

Targeting cancer cells requires innovative strategies that address their intrinsic heterogeneity and plasticity. Traditional therapies often fail because cells exist in a dynamic equilibrium, transitioning between distinct functional states, such as DTP and CSC, under specific conditions. These shifts allow cancer cells to evade treatment, leading to tumor persistence and relapse. To effectively target this adaptive system, therapeutic approaches must block the mechanisms that enable cellular plasticity, such as preventing non-CSCs from acquiring stem-like properties and restoring the CSC pool. An additional strategy could be to block the regrowth potential of DTPs (Figure 2 and Table 2).

### 4.1. Therapeutic Targeting of Cellular Plasticity and Cell-State Transitions

Targeting molecular pathways that mediate cellular plasticity, such as the transition from dormancy to proliferation in CSCs, has emerged as a viable therapeutic strategy to prevent tumor relapse [25]. In CRC models, dormant LGR5^+^p27^+^ CSCs survive chemotherapy and later reinitiate tumor growth via chemotherapy-induced disruption of the COL17A1–FAK–YAP signaling axis. Pharmacological inhibition of FAK or YAP using small-molecule FAK inhibitors, TEAD inhibitors, or inducible YAP/TAZ knockdown effectively suppressed dormancy exit and delayed tumor regrowth in both organoid-based relapse models and xenograft assays. These results support the use of chemotherapy in combination with agents targeting dormancy-breaking signals to disrupt CSC plasticity and thereby prevent recurrence.

The study by Álvarez-Varela et al. [26] provides further evidence to support that targeting cellular plasticity can improve treatment outcomes in CRC. The authors identified slow-cycling DTP cells marked by MEX3A, which emerged under suboptimal niche conditions. In patient-derived organoids and mouse models, MEX3A^+^ cells survived standard chemotherapeutic regimens (FOLFIRI/FOLFOX) and regenerated tumors after treatment. Genetic or conditional ablation (inducible caspase-9 system) of MEX3A^+^ cells impaired plasticity, promoted differentiation, increased chemosensitivity, and delayed tumor regrowth in xenograft models. These findings identify MEX3A^+^ DTP cells as a key driver of resistance and support targeting plastic states to improve CRC treatment durability.

Similarly, Coppo et al. [27] identified a slow-growing, drug-resistant subpopulation (S-cells) within PDOs that survived treatment with 5-FU and the MEK inhibitor, PD0325901. Upon drug withdrawal, these S-cells transitioned into fast-proliferating cells via a Notch/MSI1-dependent mechanism. Importantly, pharmacological inhibition of Notch signaling using the γ-secretase inhibitor DAPT effectively blocked this transition and suppressed tumor regrowth. These findings suggest that blocking cell growth transition, rather than targeting proliferation alone, may prevent relapse. Further, these findings highlight Notch/MSI1 as a promising therapeutic target.

Calandrini et al. [71] showed that Wilms tumor organoids retain epithelial, stromal, and blastemal-like cell populations, with cell-state heterogeneity linked to drug sensitivity. Organoids derived from pretreated tumors showed reduced sensitivity to vincristine, suggesting therapy-induced selection of resistant states. Panobinostat, a pan-HDAC inhibitor, selectively killed tumor organoids compared with normal kidney controls, suggesting the presence of an epigenetic vulnerability linked to tumor plasticity. Blastemal-like organoids were particularly sensitive to etoposide and the p53-activator idasanutlin, revealing state-specific therapeutic targets. These results support the targeting of plasticity using epigenetic or state-targeted approaches.

### 4.2. Targeting Niche-Dependent Cellular Plasticity and Cell States

Plastic tumor cell states are often sustained by niche-derived signals, offering therapeutic windows before the cells transition to autonomous, therapy-resistant phenotypes. Seino et al. [29] showed that Wnt–non-secreting (W−) PDAC organoids rely on CAF-derived Wnt ligands and are vulnerable to porcupine inhibitors (Porcn-i), which block Wnt secretion and suppress tumor growth in vitro and in xenografts, regardless of RNF43 status (RNF43 loss sensitizes tumors to Wnt ligands by increasing frizzled receptor stability). Co-culture with CAFs promoted W− organoid growth via juxtacrine Wnt signaling, which was reversed by Porcn-i. These findings suggest that targeting stromal Wnt signaling can eliminate niche-dependent tumor cells before they acquire resistance.

Yan et al. [70] used CRC organoids to demonstrate that Wnt pathway dependencies can guide strategies targeting plasticity. APC–wild-type tumors with PTPRK–RSPO3 fusions or RNF43 mutations were Wnt-dependent, differentiated upon Wnt withdrawal, and responded to porcupine inhibitors (IWP2 and LGK974) and anti-RSPO3 antibodies. In contrast, APC-mutant organoids showed constitutive Wnt activity, remained undifferentiated, and expressed high levels of PTK7. A PTK7-targeted antibody–drug conjugate (PF-06647020) showed promise in preclinical trials [73], and a clinical trial is currently ongoing (NCT04189614).

Transcriptional plasticity in pancreatic cancer cells, driven by microenvironmental signals such as TGF-β and IFN-γ, can dynamically modulate drug sensitivity. Exposure to these factors induces a shift toward basal-like or intermediate co-expressor (IC) states, which are associated with reduced responsiveness to standard chemotherapies, but increased sensitivity to MAPK pathway inhibitors. These state transitions are reversible, highlighting the potential of targeting cell-state plasticity as a therapeutic strategy for overcoming resistance in PDAC organoid models [31]. Additionally, plasticity can be exploited not only by targeting state-specific dependencies but also through sequential or alternating drug regimens to harness evolutionary dynamics and delay resistance [74].

### 4.3. Potential Therapeutic Targets

Cellular plasticity not only drives intratumor heterogeneity but also represents a critical mechanism of therapy resistance, making it a promising therapeutic target. Togasaki et al. [35] found that the plasticity between PCC-NOS and SRCC in diffuse gastric cancer is regulated by extrinsic Wnt/R-spondin signaling and is enhanced by TP53 mutations. This suggests that targeting Wnt production may promote differentiation and suppress tumor growth, especially in TP53-mutant tumors, where plasticity fuels tumor maintenance and therapy resistance. In SCNEC/cervix organoids, chemotherapy induces a reversible neuroendocrine-to-adenocarcinoma shift without loss of viability, indicating that lineage transdifferentiation is a resistance mechanism [34]. Miyabayashi et al. [30] showed that RAS-driven plasticity via EMT and TGFβ signaling promotes transition to basal-like PDAC, and that targeting RAS-regulated surface and secreted proteins may help maintain less aggressive states.

In CRC models, targeting LGR5^+^ CSCs alone is insufficient, as LGR5^−^ cells can revert to a stem-like state and drive tumor regrowth [23,24]. Shimokawa et al. [24] enhanced therapeutic efficacy by combining short-term CSC ablation with cetuximab, achieving >95% tumor regression in some xenografts, an effect not observed with monotherapies or oxaliplatin, possibly because of their failure to induce LGR5 expression. These findings support the use of rational combinations that target CSC plasticity. Similarly, Fumagalli et al. [36] demonstrated that LGR5^−^ cells seed metastases but require reversion to LGR5^+^ CSCs for outgrowth, underscoring the need to co-target cellular plasticity to prevent metastatic progression. Fibroblast-derived signals in the tumor microenvironment can drive CRC cells into a slow-cycling revival CSC (revCSC) state via YAP activation [32]. This revCSC population is functionally linked to chemoresistance and resembles DTP cells. In contrast, oncogenic mutations, such as APC loss and KRAS^G12D, favor a hyperproliferative proCSC fate. Notably, the combination of YAP inhibition (to block access to the revCSC state) with standard chemotherapies (to target proliferative proCSCs) has been proposed as a potential therapeutic strategy for eliminating both plastic and proliferative-resistant cell populations. A recent study revealed that CAFs changed PDOs from proliferative to “revival” stem-cell states with TGF-β/YAP signaling, resulting in chemoprotection, and YAP inhibition reversed CAF-induced nuclear YAP and restored sensitivity to chemotherapy [72]. Targeting such cell-state transitions may be a novel strategy to overcome resistance and improve treatment outcomes in CRC.

## 5. Conclusions

Cancer cell plasticity represents a paradigm shift, redefining cancer cells as dynamic entities that drive tumor evolution and therapeutic resistance. This challenges traditional views and poses new questions: are plastic cells rare, or can most tumor cells adopt adaptive states under stress, such as during therapy? Targeting this “moving target” highlights the limitations of conventional treatments that overlook the unique properties of plastic cells. Integrating novel insights into cancer cell plasticity offers insights into new therapeutic opportunities. Future research should prioritize the development of therapeutic agents that specifically inhibit these adaptive pathways by focusing on the mechanisms that enable cancer cells to switch between different functional states and evade conventional therapies. Additionally, improving model systems, such as PDOs and single-cell assays, is essential to enable better studies on plasticity and screening of targeted therapies. The disruption of these adaptive mechanisms in cancer cells may lead to more durable therapeutic responses.

Research on plasticity based on PDO drives the development of treatment strategies and precision in cancer medicine. Examples include new combinations of drugs with chemotherapy and the discovery of novel targets in stromal cues. Furthermore, integrating the PDO platform into precision cancer medicine has the potential to aid patient stratification [75,76], recurrence prediction [25,26], and the design of personalized therapies [31,32].

Although PDOs represent a powerful platform for modeling intratumor heterogeneity and cancer cell plasticity, they have some limitations. Established organoids lack stromal, vascular, and immune compartments. This absence restricts the ability of PDOs to fully capture tumor–microenvironment interactions. Recent efforts to overcome these limitations include co-culture systems with fibroblasts, endothelial cells, and immune cells, as well as organoid-on-chip platforms that incorporate vascular flow and extracellular matrix dynamics [77,78,79]. These approaches have the potential to narrow the gap between PDOs and tumors in vivo; however, the culture system itself is essentially an artificial model; therefore, caution is still required when interpreting data obtained from the co-culture system.

## Figures and Tables

**Figure 1 cells-14-01464-f001:**
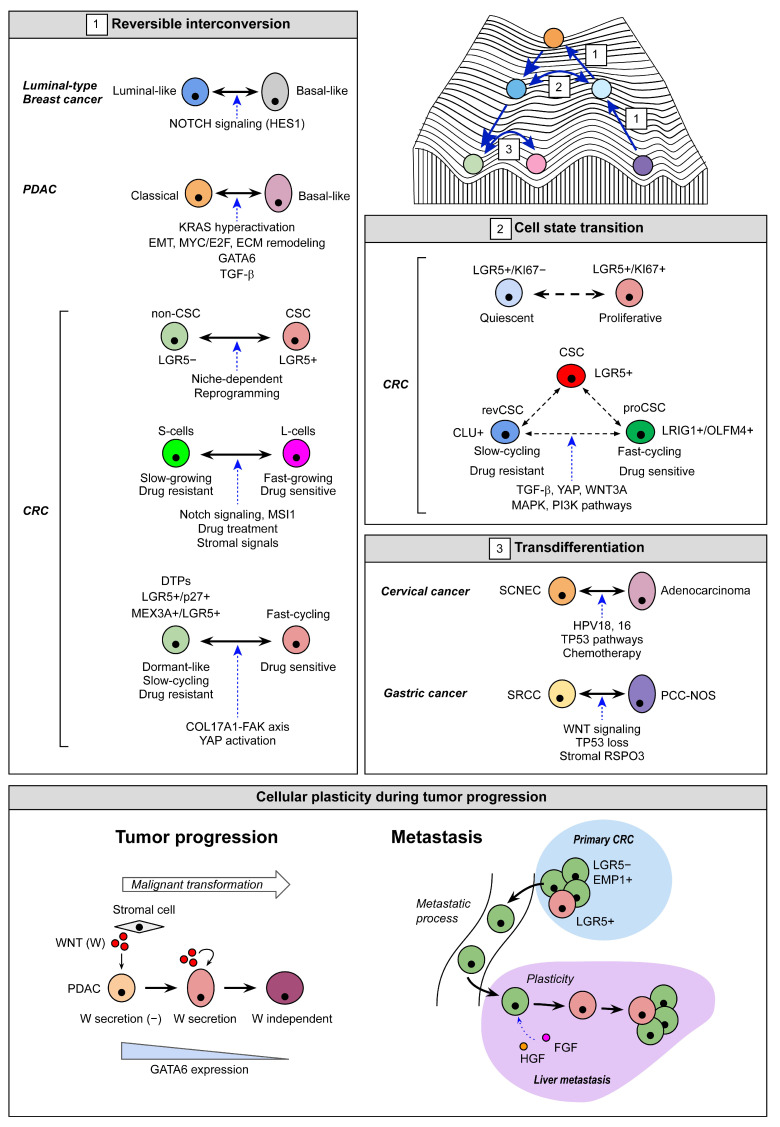
Cellular plasticity in cancer and its modeling using organoids. Schematic summary of key plasticity mechanisms across cancers. (**1**) Reversible interconversion: tumor cells transition bidirectionally between phenotypic states (e.g., non-cancer stem cell [CSC]—CSC, slow—fast-cycling) in breast, pancreatic, and colorectal cancers, driven by signaling pathways and microenvironmental cues [23,24,25,26,27,28,29,30,31]. (**2**) Cell state transitions: stem cells exist in dynamic states (e.g., revival CSCs [revCSCs] and proliferative CSCs [proCSCs]) that interconvert under niche signals [32,33]. (**3**) Transdifferentiation: lineage switching occurs in cervical and gastric cancers through oncogenic signaling and environmental inputs [34,35]. Bottom panel: during tumor progression and metastasis, plasticity enables adaptation to new niches (e.g., liver), supporting survival and outgrowth. Organoid-based approaches enable mechanistic study of these transitions [29,36,37]. In the Waddington model, different cellular states are represented by different colors, and each transition direction between cellular states is indicated by an arrow and a corresponding number.

**Figure 2 cells-14-01464-f002:**
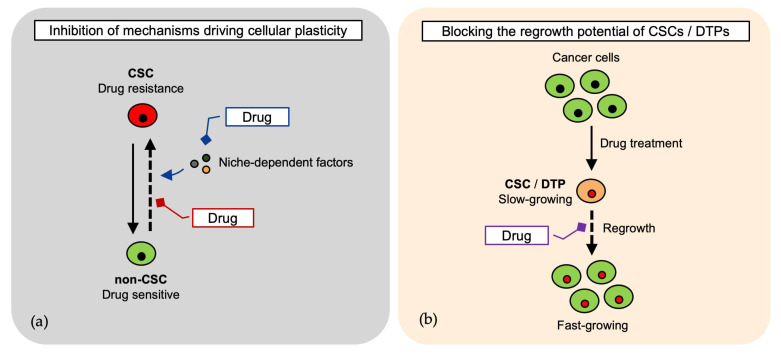
Therapeutic strategies targeting cancer cell plasticity and state transitions. (**a**) Cancer cells transition between distinct cancer stem cell (CSC) states, contributing to therapy resistance and relapse. Blocking plasticity mechanisms—such as niche-dependent cues and dedifferentiation of non-CSCs—may prevent CSC pool restoration [24,26,29,31,32,35,36,70,71,72]; (**b**) Targeting the regrowth potential of drug-tolerant persister (DTP) cells and CSCs to inhibit tumor persistence [25,26,27,32].

**Table 2 cells-14-01464-t002:** Organoid-derived therapeutic targets of cancer cell plasticity.

Tumor Type	Strategy	Target/Pathway	Therapeutic Agents	Reference
Wilms Tumor (Kidney)	Exploit plasticity-linked vulnerabilities in cell states	Epigenetic modulation (e.g., HDAC)	Panobinostat, etoposide, idasanutlin	[71]
Gastric Cancer	Suppress plasticity and promote differentiation by targeting niche signals	Wnt/R-spondin and TP53 mutation axis	Wnt ligand inhibitors	[35]
PDAC	Disrupt stromal Wnt support for niche-dependent tumor cells	Porcupine-Wnt signaling	Porcn inhibitors (e.g., LGK974), anti-RSPO3 antibodies	[29]
	Target reversible state transitions modulating drug sensitivity	TGF-β/IFN-γ-driven transcriptional states	MAPK pathway inhibitors	[31]
Colorectal Cancer (CRC)	Combine CSC ablation with chemotherapy (e.g., cetuximab)	LGR5^+^ CSCs and plasticity	Anti-EGFR antibody (cetuximab)	[24]
	Inhibit FAK/YAP to prevent dormancy exit	COL17A1-FAK-YAP	FAK inhibitors, TEAD inhibitors, inducible YAP/TAZ knockdown	[25]
	Ablation of MEX3A to block adaptive plasticity and sensitize to chemotherapy	MEX3A^+^ DTP cells	N/A	[26]
	Block S-cell to-regrowth transition via Notch inhibition	Notch/MSI1	γ-secretase inhibitor (DAPT)	[27]
	Dual targeting of YAP-induced revCSCs and PI3K/MAPK-driven proCSCs	revCSCs via YAP/TGFβ/WNT3A	YAP inhibitors + chemotherapy	[32]
	Exploit differentiation capacity or target Wnt surface molecules	Wnt-dependent vs. autonomous Wnt signaling	LGK974, anti-RSPO3, PF-06647020 (discontinued)	[70]

PDAC, pancreatic ductal adenocarcinoma; CSC, cancer stem cell; DTP, drug-tolerant persister cell; S-cell, slow-growing cell; revCSC, revival stem cell; proCSC, proliferative stem cell.

## Data Availability

No new data were created or analyzed in this study. Data sharing is not applicable to this article.

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
