# Peer review of "Patient-Derived Tumor Organoids to Model Cancer Cell Plasticity and Overcome Therapeutic Resistance"

_cells, 2025, doi:10.3390/cells14181464_

Round 1

Reviewer 1 Report

Comments and Suggestions for Authors

In this review manuscript entitled "Patient-Derived Tumor Organoids to Model Cancer Cell Plasticity and Overcome Therapeutic Resistance”, the authors discuss the new insights derived from the emerging approach, tumor organoid modeling, and its potential benefits in understanding the intratumoral heterogeneity of the tumor cells and their plasticity in conjunction with the tumor microenvironment. The authors also describe that the tumor organoid modeling can provide useful knowledge with translational value for overcoming cancer therapeutic resistance. This is a timely topic, which is of interest for the broad readership of the journal.

There are, however, a few things that the authors can address to increase the quality of the mannuscript.

1) The patient-derived tumor organoids (PDOs) can better recapitulate the intratumoral heterogeneity than 2D culture. However, some of the critical tumor microenvironment elements might be missing in many cases, such as blood vessele, nervous system, immune cells, etc. It would be nice if the authors can discuss the limitation of the PDOs models.

2) Although the authors set subsections in "3 Organoids to Model Cellular Plasticity, Cell Fate, and Cell State Transitions", the difference between Reversible interconvertion vs Cell state transition is not clear and the discussions for both seem to be overlapping.

3) Can CRC organoid models discussed in the section 3 be consolidated into section 4, which is to cover CRC related topics? Or, can other CRC models discussed in section 4 be discussed in section 3 according to the biological topic to be disscussed?

4) The majority of the information discussed in this review, even outside the section 4, is mostly CRC organoids. As such, the authors may want to state in the abstract that the discussion is mostly based on CRC organoids, if possible, with reasoning. It would be also worth discussing a report on a systematic assessment on drug response and TME utilizing CRC PDOs [https://doi.org/10.1016/j.cell.2023.11.005].

5) Page 4, lines 156-159 "In the absence of inflammation, only stem cells form tumors after APC deletion. However, co-deletion of APC and NFKBIA activates NF-κB signaling and enables non-stem cells to initiate tumors, highlighting the role of environmental signals in promoting tumor-initiating potential [29–31]."

In the way these observations are described in the text, it is difficult to appreciate the role of environmental signals that promote tumor initiating potential. It sounds rather that genetic alteration plays a critical role for that process.

6) Page 14, lines 527-529, "A PTK7-targeted antibody–drug conjugate (PF-06647020) showed promise in preclinical trials, although its clinical development was discontinued after Phase I (NCT04189614)."

Was this clinical trial discontinued due to adverse effects or no expected effects? Either way, this can be an example indicating a limitation of organoid-based approach.

7) Page 14, lines 531-536, "Exposure to these factors induces a shift toward basal-like or intermediate co-expressor (IC) states, which are associated with reduced responsiveness to standard chemotherapies but increased sensitivity to MAPK pathway inhibitors. These state transitions are reversible, highlighting the potential of targeting cell-state plasticity as a therapeutic strategy for overcoming resistance in PDAC organoid models [57]."

This is a very intriguing reaction of the tumor cells that can be also exploited by multi-drug alteration adaptive therapy [https://doi.org/10.3390/cancers14112699].

Author Response

We thank the reviewer 1 for the thoughtful evaluation of our manuscript. The constructive feedback has helped us substantially improve the manuscript. Detailed, point-by-point responses are provided in the attached PDF file. All changes in the manuscript are marked in blue for easy reference.

Reviewer 2 Report

Comments and Suggestions for Authors

This study reviewed PDOs to study cancer cell plasticity, with emphasis on the dynamic behavior of cancer stem cell (CSC) and DTP populations and emerging strategies to target plasticity-driven resistance. There ar some potential points that can improve the current version;

  • The novelty of this study should be clearly highlighted in comparison with recent publications, such as Mohammad Hadi Abbasian et al., 2025.

  • The clinical relevance and potential applications of the reviewed content should be more strongly emphasized.

Author Response

We thank the reviewer 2 for the thoughtful evaluation of our manuscript. The constructive feedback has helped us substantially improve the manuscript. Detailed, point-by-point responses are provided in the attached PDF file. All changes in the manuscript are marked in blue for easy reference. We improved the quality of English language through English editing service (Editage).

Reviewer 3 Report

Comments and Suggestions for Authors

Brief summary:

In this review, Coppo and Inoue have provided an overview of cancer cell plasticity and how organoid models can be used to study the mechanisms of how it can arise. Cancer cell plasticity refers to the ability of cancer cells to transition between different cell states, including more stem-like and drug resistant states, without genetic mutations. This review categorises cell plasticity into three major mechanisms: reversible interconversion, cell state transition, and transdifferentiation. It also describes how these mechanisms can promote tumor growth and metastasis. At the same time, it also describes various organoid models that recapitulate these mechanisms of cell plasticities, and potential therapeutic approaches to target these cells.

General comments:

- The article provides a clear overview and understanding of various mechanisms of cell plasticity, and adequate references for each of these examples. The topic under review is of relevance with the current understanding of cancer cell plasticity, and has a well-defined scope of organoid models for studying this phenomenon.
- This review may benefit from a short discussion of cellular transdifferentiation and EMT in a non-cancer context, which have been recognized in development, wound healing and tissue regeneration, to be consistent with how other mechanisms are also discussed in a non-cancer context.
- It would be very useful to have a quick summary of the different organoid culture/establishment methods discussed in this review, as the papers discussed here spans a range of techniques including PDO cultures, PDX cultures, and in vivo implantations, as well as different ways of maintaining the organoids in culture.
- There is some repetitiveness in the organization, where a given study is discussed several times in different sections.

Specific comments:

Introduction
- Please be more specific in the statement of "improved in vivo relevance" (line 38)
- Please be more specific about "niche factor dependence" (line 44)

Figure 1
- The Waddington landscape is very helpful in distinguish between the different mechanisms of plasticity. The caption description should make a bigger distinction between reversible interconversion and cell state transitions.
- If possible, please indicate which direction the listed pathways influence the cell state

Table 1
- Clarification or renaming of the "Mechanism/Pathway" column would be useful in interpreting the table. In some references it indicated mechanisms that are involved in cellular plasticity, while other references it describes the cell states ("Stem-like but non-proliferative") or other observations about the system that is not involved in effecting cell plasticity ("immune vulnerability window").
- (Also for Figure 1) Reference 54 within the breast cancer section does not show transition between luminal and basal subtypes based on the indicated pathways. The experiments in the paper show the ability of epithelial progenitors of each subtype to re-generate the other subtypes present in the mammary gland over the long culture time. The pathways indicated are only shown to affect the proportion of different subtypes, which most likely affects relative proliferation instead of cell type plasticity. Also "lineage marker profiling" is not a mechanism.

Chapter 3.1
- I wouldn't say ER is a hallmark of luminal cell identity. In the mammary gland there are two major subtypes of luminal epithelial cells, one of which is ER-negative. Even among the hormone receptor-positive luminal cell population, the rate of ER expression is about 55.4% (https://www.nature.com/articles/nm.2000/tables/1). However, it is true that ER expression is a key functional feature of the hormone receptor-positive luminal epithelial cells.

Chapter 4
- The descriptions of studies referenced by 64 and 65 share many commonalities and could be discuss at the same time.

Figure 2
- It might make it more clear to reference the drugs targeting cellular plasticity with a different name compared to the intiial therapeutic treatments.

Table 2
- I think the flow of reading would work better if the "Strategy" column is swapped with the "Target/Pathway" column.

Chapter 5.3
- Separating out this chapter from 5.1 and 5.2 didn't seem necessary. Some of the potential therapeutic approaches seem like they will fit in with the other chapters. For instance, the section about targeting YAP pathway and standard chemotherapies in CRC could be in chapter 5.1.

Author Response

We thank the reviewer 3 for the thoughtful evaluation of our manuscript. The constructive feedback has helped us substantially improve the manuscript. Detailed, point-by-point responses are provided in the attached PDF file. All changes in the manuscript are marked in blue for easy reference.
